# Features of Mobile Apps for People with Autism in a Post COVID-19 Scenario: Current Status and Recommendations for Apps Using AI

**DOI:** 10.3390/diagnostics11101923

**Published:** 2021-10-17

**Authors:** Ikram Ur Rehman, Drishty Sobnath, Moustafa M. Nasralla, Maria Winnett, Aamir Anwar, Waqar Asif, Hafiz Husnain Raza Sherazi

**Affiliations:** 1School of Computing and Engineering, University of West London, London W5 5RF, UK; ikram.rehman@uwl.ac.uk (I.U.R.); maria.winnett@uwl.ac.uk (M.W.); 21452391@student.uwl.ac.uk (A.A.); waqar.asif@uwl.ac.uk (W.A.); 2Faculty of Business, Law and Digital Technologies, Solent University, Southampton SO14 0YN, UK; drishty.sobnath@solent.ac.uk; 3Department of Communications and Networks Engineering, Prince Sultan University, Riyadh 11586, Saudi Arabia; mnasralla@psu.edu.sa

**Keywords:** autism, special educational needs, mobile apps, COVID-19, augmentative and alternative communication, applied behaviour analysis, artificial intelligence

## Abstract

The new ‘normal’ defined during the COVID-19 pandemic has forced us to re-assess how people with special needs thrive in these unprecedented conditions, such as those with Autism Spectrum Disorder (ASD). These changing/challenging conditions have instigated us to revisit the usage of telehealth services to improve the quality of life for people with ASD. This study aims to identify mobile applications that suit the needs of such individuals. This work focuses on identifying features of a number of highly-rated mobile applications (apps) that are designed to assist people with ASD, specifically those features that use Artificial Intelligence (AI) technologies. In this study, 250 mobile apps have been retrieved using keywords such as autism, autism AI, and autistic. Among 250 apps, 46 were identified after filtering out irrelevant apps based on defined elimination criteria such as ASD common users, medical staff, and non-medically trained people interacting with people with ASD. In order to review common functionalities and features, 25 apps were downloaded and analysed based on eye tracking, facial expression analysis, use of 3D cartoons, haptic feedback, engaging interface, text-to-speech, use of Applied Behaviour Analysis therapy, Augmentative and Alternative Communication techniques, among others were also deconstructed. As a result, software developers and healthcare professionals can consider the identified features in designing future support tools for autistic people. This study hypothesises that by studying these current features, further recommendations of how existing applications for ASD people could be enhanced using AI for (1) progress tracking, (2) personalised content delivery, (3) automated reasoning, (4) image recognition, and (5) Natural Language Processing (NLP). This paper follows the PRISMA methodology, which involves a set of recommendations for reporting systematic reviews and meta-analyses.

## 1. Introduction

COVID-19 has had an unprecedented impact on life around the world. Among all, people with Autism Spectrum Disorder (ASD) are the ones that are heavily affected. ASD is a neuro-developmental disability that permanently affects social communication, inhibits the development of appropriate peer relationships and includes repetitive patterns of behaviour or activities [1]. It is among the most enigmatic child development disorders with a dramatic increase in prevalence, with one in a hundred children in the UK having this disability. An estimate reveals that 700,000 people in the UK are diagnosed with autism as of 2020 [2].

According to the Mental Health Foundation, as well as the NHS [2,3], people with ASD typically exhibit one or more of the following characteristics:

### 1.1. Verbal and Non-Verbal Communication

People with ASD are often delayed in learning to speak, and both children and adults with ASD tend to find it difficult to express how they are feeling. They tend to have a very literal understanding of language and have difficulty understanding metaphors, sarcasm, idiomatic expressions and euphemisms. Problems with non-verbal communication include avoiding eye contact and circumventing the use of facial expressions to express sentiments. However, this does not necessarily mean that they are not able to communicate, but rather they opt for other non-conventional means to relay their thoughts.

### 1.2. Social Understanding and Social Behaviour

People with ASD often have difficulties in understanding social behaviours. They tend to avoid taking part in conversations and prefer to be alone. They portray an image of being blunt, rude and uninterested in a situation with occasional interruptions; they end up talking over others. They also tend to have difficulties empathising with others, particularly in social situations, making it hard for them to make friends. This can become a difficult situation for children, making it harder for them to play with others.

### 1.3. Inflexible Behaviour and a Lack of Imagination

People with ASD tend to exhibit repetitive behaviour such as persistent hand gestures or repetitive body rocking. They tend to prefer a specific routine and may get very upset if changes are implicated. They are also known to exhibit a lack of imagination. This can often lead to an increase in preference for solitary activities and make them very anxious in social situations. However, they can exhibit an excellent memory in their particular interest, making them very passionate and obsessive about them.

The consequences of ASD are profound, both for people with ASD as individuals and society as a whole. The characteristics of those with ASD discussed above typically lead to social isolation and learning difficulties, making it harder for them to learn than most of their peers. Schools play a vital role in ensuring pupils with autism have access to the bespoke curriculum along with the therapeutic support from the specialists for them to engage in their learning. These students benefit from structured content with little to no changes in their routine. In a pre-COVID-19 setting, this translated to following norms of physically associating and coming in close contact with others outside the child’s family. This highlighted the difficulties pupils had and, thus, helped in the diagnosis of ASD at an early age.

Moreover, having scheduled social contact with a trained autism specialist helps pupils with autism earn life skills targeted, yielding increased independence at home, at school and in the community [4]. Despite these practices, the statistics from the Centers for Disease Control and Prevention (CDC), United States, indicate that 1 out of 68 children are diagnosed with ASD, out of which 20–30% could not communicate their needs, wishes and thoughts [5]. This ability had a severe impact on the individuals’ Quality of Life (QoL). In contrast, they had the least opportunity to gain access to education, particularly in developing countries, where they faced numerous other problems such as depression and social anxiety [6]. Moreover, only about 6% to 6.7% of people with a learning disability are in paid work in England and Scotland, according to the statistics provided by the UK government [3]. A deficit of appropriate work opportunities and career guidance affect young people with ASD and subsequently leads to unemployment. These figures have been disproportionately affected by the pandemic [4].

The rules governing social distancing dramatically changed social interaction, and this helped highlight the deficiencies in the current system. On one side, social distancing helped alleviate the necessity of interacting with others outside one’s family. On the other hand, the imminent danger of creating socially excluded individuals has increased [7]. Moreover, the disruption of critical services along with changes in daily lives has adversely affected the autistic symptoms of people with ASD, thus increasing behavioural challenges and declining the level of mental well-being [8]. Furthermore, the pre-COVID-19 conventional approaches resulted in most children under 4 years not being diagnosed properly, with about 27% of children remaining undiagnosed by the age of 8 [9]. While early intervention is essential so as to maximise a child’s learning potential while the brain is still developing, children and families have to wait months or years before receiving a diagnosis, a problem exacerbated by the shortage of resources, the constantly changing social interaction protocols and the lack of appropriate telehealth solutions [10]. Numerous questionnaires have been used to detect various signs of autism, such as the Modified Checklist for Autism (M-CHAT) [11] or the Child Behavior Checklist (CBCL) [12]. However, these tools still need to be interpreted by experts, usually medical practitioners who might not be very accurate for individuals on the mild end of the spectrum.

Researchers have already started looking into questions such as: How can advances in technology assist those with ASD and improve their quality of life? Moreover, how can technologies assist those interacting with people with ASD? While mobile phone penetration is forecasted to grow in the next few years, mobile phones can be considered an excellent platform that provides ubiquitous computing. The functionality of mobile devices can be enhanced to improve the lives of those with ASD; this can provide an effective and low-cost solution. As evidence of the ubiquity of mobile apps, there has been a rapid increase in the number of mobile app users in the last decade, with a total of 2.1 million apps available on the Google Play Store for Android users and almost 2 million mobile apps available to download from the Apple App Store by the year 2018 [13]. Providing ASD candidates access to the latest technologies has proven improvement in developing specific skill sets [13,14]. Such technologies include facial expression recognition, video games, and character animation, which have been shown to be successful for teaching students with autism [15,16].

In this paper, we reviewed mobile applications for individuals with ASD, discussing the state-of-the-art solutions and approaches in this scope. In order to provide a better understanding of the conducted work, we propose a methodology for selecting apps using sentiment analysis and apps rating, specifically those apps that use AI technologies. Furthermore, 250 mobile apps have been retrieved using keywords such as autism, autism AI, and autistic. Among 250 apps, 46 were identified after filtering out irrelevant apps based on defined elimination criteria such as ASD common users, medical staff, and non-medically trained people interacting with people with ASD. Following the elimination criteria, 25 apps were downloaded for analysis. These apps were based on features such as eye tracking, facial expression recognition, 3D cartoons, haptic feedback, etc. In addition, recommendations for software developers and healthcare professionals are provided in designing AI-based apps to cater for autistic people. The work presented in this paper is novel as it addresses an important topic in the domain of special needs education and lifestyle management. To the best of the authors’ knowledge, no similar feature analysis of available mobile apps aimed at assisting those with ASD currently exists. Furthermore, no existing studies discuss the potential impact of AI on mobile app development to help people with ASD.

This study has the following threefold contributions:1.The existing mobile apps are analysed based on a set of common features developed for people with ASD.2.A methodology is proposed for selecting apps using sentiment analysis and app ratings.3.Future AI-based recommendations are provided for ASD application development.

The rest of this paper is organised as follows: Section 2 covers the literature review discussing the state-of-the-art on existing apps and their capabilities to address the needs of the people with ASD. The methodology, including selection criteria and user reviews sentiment analysis, is presented in Section 3. Moreover, Section 4 analyses selected mobile applications aimed at users with ASD and the effectiveness of their presented features in addressing the needs of people with ASD, specifically exploiting the usage of Artificial Intelligence (AI) technologies. Similarly, Section 5 discusses the effectiveness of using AI technologies to be adopted by the software developers and research scholars to achieve further improvements in mobile apps that can assist both users with ASD and those they interact with. Furthermore, Section 6 provides concluding remarks and recommendations. Finally, a set of limitations and future research potentials are unfolded in Section 7.

## 2. Related Work

For the systematic literature review (SLR), we collected and analysed research articles and technical reports using the keywords such as ‘Autism Spectrum Disorder (ASD)’, ‘ASD Post COVID-19’, AI/ML approaches for detection of ASD’, ‘ASD and E-Learning’, ‘Mobile Apps for ASD’, ASD Students’, and ‘ASD in Employment’. These keywords resulted in articles that were published in widely used electronic databases (e.g., IEEE Explore, Google scholar, Science Direct, etc.) between the year 2001 to 2021. After carefully following the SLR selection criteria, 52 articles and technical reports were selected and hence have been referred to in this paper. Furthermore, the selected articles provide a rationale on the topic of mobile apps for people with Autism Spectrum disorder, as well as the relevance of these apps in a post-COVID-19 scenario.

Furthermore, the literature review also highlights the notion of the three common issues, which have been taken into account when developing apps for people with ASD, namely: (1) social interactions; (2) repetitive behaviour; (3) verbal and non-verbal interactions [17,18].

In order to enhance the learning and communication skills of ASD children, apps identified in the literature applied varying techniques, including the use of diverse sets of words, pictures, and sounds. For example, the MyTalkTools [19] mobile app helps children with ASD to communicate by expressing needs made through pictures in sentences.

Learn with Rufus [20] is another mobile app designed for children to teach them about emotions through various facial expressions, as well as to develop their language and social skills. Additionally, AutisMate [21] consists of a visual scheduling feature and sentence formation functionality. It enables parents to set up situations to prompt children to complete tasks such as washing their hands by tapping on the phone and consequently receiving rewards. Another app called the E- Mintza [22] enables users to upload photos of family members in order to allow autistic children to use these photos and make sentences, should they wish to communicate with a family member. In Bangladesh, the AutismExpress [23] app was developed to reach young adults in rural areas to assist ASD children in expressing their emotions and letting other people know about their needs through the app’s features. Furthermore, it allows parents to track other factors, including diet, daily routines, sleep cycles etc.

Furthermore, Birdhouse is another iPhone app that was developed to help parent’s self-manage their children effectively when at home by enabling them to log details of their child’s daily activities, which in turn makes it easier to establish behavioural patterns [24]. In addition, the Proloquo2Go mobile app offers multiple levels of communication, which is tailored to support a range of users, from those who cannot verbalise at all to those who simply need help in completing a sentence [25]. It also provides support for language development and can be easily customised.

In addition to these mobile apps, other technologies exist, which currently assist those with ASD. For example, automatic transcription of text can be activated on YouTube videos via the caption option, enabling students to keep track of anything their peers are discussing. The teacher’s verbal discussion about a topic can also be automatically converted to text and displayed on the screen to make students with special educational needs feel included; thus, they can fully participate in class discussions and ask questions.

Healthcare professionals have also used several existing apps to help in diagnosing people with learning disabilities [9,26]. For example, a mobile app named Cognoa assesses ASD child development in two parts: (1) a parent questionnaire having 15 items, and (2) a short video observation captured at home on the parent’s phone. The algorithms used can then classify the severity of the child’s ASD based on the gathered information. The system has shown an accuracy rate of 71% within children aged between 18–72 months [27]. In addition, using Cognoa as a mobile-health (m-health) screening tool has shown certain advantages such as efficacy, responsiveness and improved identification of ASD risks over traditional pen-and-paper screening methods.

This paper focuses specifically on areas of AI, which have been used to create applications to assist people with ASD. The smart tutoring model or virtual learning environment, for example, consists of e-learning strategies developed to define the mood and concentration level of students [28], which can later be used to understand their performance. Researchers have also utilised robotics to assist users with ASD. For example, Keepon, a robot that responds to touch and dances to music was created to engage autistic students. The robot was equipped with a camera for facial recognition and touch sensors, and autonomous teleoperation modes, which therapists remotely controlled to monitor autistic students’ interactions in a playroom [29]. Another team from the University of Pisa developed a humanoid robot called FACE (Facial Automation for Conveying Emotions), which provides information about facial expression responses to help study people with ASD. The robot also acts as a therapist to help autistic students develop social skills [29].

In 2016, Professor Einstein Robot was developed by Hanson Robotics, a Hong Kong-based engineering robotics company. It is an expressive, playful robot that helps to train the brain and can teach young people science [30]. It also offers educational games and its own dedicated mobile app Stein-O-Matic, publicly available to download from the Google Play Store and Apple’s App Store. A handful of researchers are studying the technology in an effort to find out how effectively these robots can assist children with ASD. The same company also developed Little Sophia, a 14-inch-tall robot that can teach coding and AI to children from the age of 8 onwards [31]. Little Sophia can be used in cognitive therapy and special educational needs institutions for autistic children. Another project, LIFEisGAME, is a collaborative project developed by Portugal and UT Austin to teach people with ASD about facial expressions, particularly how to recognise facial emotions in real-time through a game that focuses on technologies such as character animation and facial expression analysis [14]. Results show that children preferred engaging with 3D cartoons and animals rather than 2D ones. In addition, the game can stimulate a sense of competition in the players to make them beat their scores.

Furthermore, the Penguin for Autism Behavioral Intervention (PABI) project aims at helping autistic children to receive Applied Behavioral Analysis (ABA) therapy, which usually costs around $60,000 per year in the US [32]. Thus, with the rapid development of robotic engineering and AI, this cost can be reduced. In addition, the therapy can become accessible to more people who need treatment. Therapy data are logged and can also be reviewed by therapists at a later stage.

The literature above highlights a few projects implementing AI or robotics systems to find ways of improving the QoL of people with ASD. From the literature, we observed that not enough research has been conducted to analyse features of mobile apps for ASD individuals as a support tool. Many features seem to have been considered in the literature but are not implemented in current mobile apps. Hence, the proposed recommended features can and should be incorporated in a single app for better monitoring, follow-up by healthcare professionals, and lifestyle management for individuals with ASD. This can lead to a balance between obtaining an improved quality of life with minimal inconvenience.

## 3. Methodology

This paper adheres to a checklist of items laid down in the PRISMA approach [33], which specifies items to include when reporting a systematic review or meta-analysis. As several items in the checklist are aimed at meta-analyses of medical studies, they have been excluded as they are not applicable to this paper. However, the following items are specifically relevant and have been used as a methodology for this paper:1.The checklist demands a structured summary in the paper’s abstract, outlining methods, limitations, conclusion and implications of key findings, as outlined in the abstract above.2.The checklist demands a rationale for the review and an explicit statement of questions being addressed, as described in the introduction above.3.The checklist demands a description of information sources, an electronic search strategy including any limits used, and the process for selection, which is laid out in this section of the paper.4.The checklist demands a summary of the main findings, including the strength of evidence for each main outcome, a general interpretation of the results in the context of other evidence, and implications for further research.

These demands are covered in Section 4 and Section 5 below.

### 3.1. Sample of Selection Apps

Addressing the third item from the checklist discussed above, namely a description of information sources and strategy, the research for this paper involved an extensive search of the mobile apps that have been developed and are available on the Google Play Store to help autistic candidates. The terms autism, autistic, and autism AI were used in the search terms of the apps identified in the online mobile store. The initial screening criteria of the above search included apps with ratings of 4 stars or more out of 5 and positive user reviews of 70% or above. This yielded a total of 250 apps at the time of selection, i.e., September 2020. The motivation for using both star ratings and user review sentiment analysis is that sometimes apps are given high star ratings, but further analysis of user reviews reveals issues that users encounter while using the app, leading to an unsatisfactory user experience. Therefore, the authors’ concern was to restrict the in-depth analysis to apps with high star ratings and high user review/comments sentiment analysis to find apps with the highest performance, usability, and effectiveness currently available.

Further screening for inclusion was then carried out to extract relevant apps to be analysed in detail, as shown in Figure 1. The selected 250 apps were further screened for inclusion based on the following points:1.A removal of duplicate apps with similar features and apps not written in English.2.The inclusion of apps that applied AI-related features in a range of categories, which were identified to be Games, Education, Medical, Communication, and Others, with Others used as an umbrella term for a small but significant number of apps made for charities, fundraising and other social applications not covered by the above categories.3.Apps that were most explicitly designed for autistic users, rather than generic apps with features that might appeal to ASD users.

On this basis, 25 relevant apps were selected for in-depth feature analysis, as shown in Figure 1.

### 3.2. Reviews Sentiment Analysis

User review analysis was conducted by following the two-step process of review identification and opinion extraction [34]. This approach is simple and is already used for mobile apps reviews. Furthermore, user review sentiment analysis helps analyse the mobile application’s strengths, weaknesses, and limitations. A multistep process methodology was used following the underlined four steps:1.Review repository creation,2.Grouping of subjective words,3.Topic ranking with respect to relevance,4.Extraction of user reviews for a topic.

#### 3.2.1. Review Repository Creation

We collected reviews of apps already selected through the defined selection criteria discussed above to create a repository. For this purpose, we used Android-Market-API, a java library used to access Google’s android market servers. Through Android-Market-API, we can easily search an android application through its ID and download reviews of that particular android application. After downloading reviews, unique IDs were assigned to each application’s reviews. Moreover, review creation time was also included in application reviews. For each application, a minimum of 25 and a maximum of 75 reviews were selected for repository creation.

#### 3.2.2. Grouping of Subjective Words

Subjective word grouping was created assuming that each application review contains words such as good, bad, not working, etc. For instance, if someone likes the application, the review will contain excellent, good or working words. We also considered that these subjective words would also be directly affected by the application’s rating; that is, if application reviews contain frequent use of good, excellent, then its rating will be better and close to 5.

#### 3.2.3. Topic Ranking with Respect to Relevance

After the selection of subjective words, we ranked the subjective words based on our interest and relevance. Without using Natural Language Processing (NLP), we were left with only using the Tokenisation technique to tokenise each subjective word and compare it to our bag of interest words. Tokenisation helps in feature extraction and eases the comparison of words without using any extensive machine learning approach.

#### 3.2.4. Extraction of User Reviews for a Topic

For each application, by using the above steps, we analysed the user’s reviews and sentiments. Each application’s reviews were analysed, and with the help of subjective words, user’s sentiments about application usage and applicability were analysed.

A total of 46 applications reviews were analysed based on the above steps, and out of 46, 25 applications were selected. The criterion for inclusion was 70 or above positive reviews for each application, and this was decided through a performance metric, as shown in Figure 2. A total of 25 mobile apps (5 apps with the best ratings from each category) were downloaded and analysed, as shown in Table 1.

## 4. Results and Discussions

The features and functionalities of highly-rated apps from the Google Play Store are identified in this study, together with the identification of AI technologies that are used in these apps. Features of the analysed apps have been categorised according to the three specific characteristics of people with ASD identified in the introduction, with an additional category for apps addressing the needs of medical and non-medical staff who interact with people with ASD.

This section discusses the main findings of this study after a feature analysis has been performed in the identified apps in Section 3. Recommendations on how AI technologies could be further explored and integrated into future apps to support people with ASD are then discussed in Section 5.

### 4.1. Verbal and Non-Verbal Communication

Among the 25 apps analysed, those which assist ASD users with communication skills include Otismo (app 3), Autism iHelp-Play (app 6), Speech Assistant AAC (app 14), LetMeTalk (app 16), Card Talk (app 17), CommBoards Lite (app 18), ABtalk (app 19) and Speech Therapy Articulation (app 20). A common feature of many of these apps is the use of Augmentative and Alternative Communication (AAC). AAC mobile apps make use of a variety of symbols or images for speech mainly designed for autistic people [35], allowing users to form sentences by selecting a sequence of these symbols and images. Utilising these apps in their daily life can be life-changing for those with verbal communication difficulties. These apps can give speech to those unable to talk and provide invaluable assistance to those on the path to speech. For example, the Card Talk app (app 17) provides simple interfaces, as shown in Figure 3, to enable ASD children in making simple sentences and expressing themselves.

From our analysis, we also discovered that predominant communication apps made use of AI technologies by providing predictive text analysis and automatic speech translations [36]. Speech-to-text technology implemented in mobile apps is becoming popular among students with special educational needs, including ASD children, and has advanced rapidly in the last few years. This technology is based on AI NLP, and it shows how speech can be used to give commands to apps for performing specific tasks such as making a phone call or sending a message via voice command [37]. This technology has also proven to be highly beneficial for students who are also affected by hearing impairment.

Other apps, particularly games, focused on helping ASD children recognise faces and interpret expressions using virtual characters. The feature analysis has also revealed some apps incorporating eye-tracking software into Virtual Reality (VR) phone headsets. These headsets monitor eye movements to see how ASD players advance through different game levels. The Prism app (refer to Figure 4) consisted of 3D worlds and colourful environments that are meant to increase the interests of ASD users. It has been proven that children with autism respond best to pictures, and they process information better visually than via the auditory channels [38].

Many educational apps explicitly developed for autistic children were for stimulating reasoning, cognitive and logical development through categories such as animals, food, colours, numbers, letters and forms, such as the Jade Autism app shown in Figure 5. Similarities found in games were that they strove to increase children’s attention span and expose them to virtual worlds so that they could develop their visual learning capabilities. Common features found in the game apps also include enabling autistic children to engage in learning activities such as finding colours and numbers, stimulating cognitive and logical reasoning. Another exciting way of engaging ASD children through mobile apps was by using haptic feedback, whereby the mobile device could vibrate in specific scenarios (e.g., moving to the next level).

Design components relevant to communication that were found common in most of the analysed apps designed for autistic children are as follows:Simple colours as opposed to bright colours on the main navigation pages.The language used throughout the apps is simple: simple bullets and short sentences instead of figures or idioms.The buttons used were very descriptive and carefully selected.The layouts used in the identified apps were consistent.

It was also found that it is a common practice to avoid using euphemisms or certain expressions, for example, passed away instead of died since these create confusion among ASD children.

### 4.2. Social Understanding and Social Behaviour

A number of apps, mainly in the categories of games, education and communication, were found to be based on Applied Behaviour Analysis (ABA), a therapeutic technique based upon the principles of learning. ABA aims to change the behaviour of social significance in terms of communication, reading social situations and minimising negative behaviours by rewarding positive behaviours. The apps that were analysed included Prism (app 5), Daily Tasks (app 8), ABA Therapy Aid (app 13), ABA Dr Omnibus (app 15), TIIMO (app 23), and Autism Connect (app 24). For example, the Daily Tasks app (app 8) was designed to improve the fine motor skills and concentration spans of children with ASD. It enables children to go through various scenarios and complete each task, such as brushing their teeth or applying shampoo during their shower (Figure 6). Furthermore, it improves eye/hand coordination by enabling ASD children to focus on simple daily tasks in an engaging and interactive story. Another app that incorporates ABA principles is DrOmnibus (app 15), which provides therapeutic tools to parents, teachers, and therapists to help children acquire new skills and potentially adopt desired behaviours.

### 4.3. Inflexible Behaviour and a Lack of Imagination

As mentioned in Section 4.1 above, a common feature of the analysed apps is consistency in screen layouts, which appeals to the need of autistic people for consistency of routine and difficulties with adapting to change. In addition, many of the games analysed were found to have been programmed such that there were repetitive levels since autistic children act well in a routine environment with predefined roles. Indeed, in general, games tend to involve repetitive behaviour while simultaneously encouraging imagination and creativity in problem-solving. Furthermore, our analysis showed that the utilisation of games, whether for entertainment or educational purposes, aimed to help autistic children attain balance, attention and gaze control.

### 4.4. Support for Medical Staff, Parents and Carers of People with ASD

Many of the apps analysed targeted medical staff, academic staff, parents and carers to assess the social attention and communication behaviours of people with ASD. In addition, teaching materials based on ABA therapy and developed by specialists, such as those mentioned in Section 4.2, provide guidance and support for carers of autistic children. Common features among the apps are short videos and links to external sources that can help informal carers develop their knowledge about autism.

There were only a few highly rated medical apps. Out of the total 46 shortlisted, only 8 were specific to medical. Some medical apps were developed to encourage Applied Behaviour Analysis (ABA) [41]. Some of the medical apps identified in the study consisted of a pre-screening questionnaire to help doctors identify any possible symptoms in patients to be referred for further evaluation. For example, the Awesomely Autistic Test (app 11) and Autism Test (app 12) have been designed for early detection of autism. They are used by doctors to assess different levels of the condition based on international guidelines for autism, such as the National Institute for Health and Care Excellence (NICE) [42] and the American Association for Child and Adolescent Psychiatry (AACAP) [43]. Additionally, medical apps shared similar functionalities to some communication apps as some of the medical-classified apps embedded speech therapy and were therefore approved as a medical device.

### 4.5. Use of AI Technologies

There were not many apps in the market that exploited AI technologies to help with self-diagnosis or to predict the performance of autistic children in their education. No apps were found that could predict skills that people with ASD are more likely to develop while playing certain games.

Figure 7 shows the common features of the identified top-rated apps considered in this paper’s feature analysis, with AI-related features highlighted in bold. In addition, the figure includes features taken from the literature, which software developers can collectively consider in designing future apps for each of the aforementioned categories.

## 5. Recommendation for Future Apps Development

Artificial Intelligence (AI) is a field of computer science concerned with the development of software to perform human-like actions such as learning, reasoning and self-correction [44], among other activities. It includes concepts such as Machine Learning (ML), NLP, and robotics, etc. These concepts are being implemented in various systems, including mobile apps, to enhance inclusion and accessibility for SEND individuals [45,46,47].

Special Educational Needs and Disabilities (SEND) refers to individuals who have learning problems or disabilities, including ASD, which make it harder for them to learn than most of their peers. AI has already started reshaping how people learn and communicate by enabling ubiquitous use of intelligent, human-centric and personalised machines and/or software anytime and anywhere. Advances in technologies and AI methods are accelerating the diagnosis and treatment of people affected by different conditions. AI has recently been applied to various types of healthcare data regarding major diseases such as cancer, neurology, and cardiology [48].

Thus, certain AI applications can also be exploited, explored, and integrated within mobile apps, which will reform support for people with ASD. Attributed advantages of deploying AI-based mobile apps include their ability for personalisation: it can be tailored towards individuals’ specific learning difficulties, thereby making provisions for self-improvement over personal learning difficulties. Personalised education can enable individuals to learn at their own pace, particularly for people with ASD who may vary from one another with different needs, making group teaching difficult [10]. AI-based systems integrated within mobile apps can facilitate teaching procedures and increase the concentration level of students, as shown by some studies such as in [6,13]. Mobile devices embed new technologies while being ubiquitous, thereby enabling people with specific disabilities to feel more connected and included in society.

With advances in AI, there are significant opportunities for developing intelligent systems to help people with ASD in their day to day tasks, either at home or at work. As a result, the features and functionalities of highly-rated apps identified in this study can be developed further by integrating AI technologies.

Future software engineers can consider such technologies when developing novel systems for people with ASD to improve their lifestyle in the area of societal inclusiveness in the near future. This section provides recommendations on how existing apps could be further developed to empower people with autism to create more meaningful experiences using AI technologies for (1) Progress Tracking; (2) Personalised Content Delivery; (3) Automated Reasoning; (4) Image Recognition, and (5) Natural Language Processing. To the best of our knowledge, the literature lacks published materials on the efficacy and features necessary to be integrated into future mobile applications developed for ASD.

Machine learning, an application of AI [47], holds the potential to detect or diagnose people with learning disabilities at an early stage. It can help clinicians in decision-making and spot a child’s susceptibility to the spectrum disorder by analysing behavioural indicators collected over time. With deep learning, which is a branch of machine learning, patterns can be recognised amongst various other features, which are then analysed by different algorithms to aid trait recognition that may be overlooked by the human eye [48]. Feature engineering is a costly but fundamental process that requires a lot of effort and time before developing a machine learning tool. It is the process of analysing and understanding the domain knowledge of big data that will need to be fed to the learning algorithm’s platform [49]. ASD patients’ complex medical data, their psychological and behavioural factors affecting the condition, are not always recorded in a standard way across hospitals, and this poses challenges for engineers trying to implement AI-based systems. Moreover, in order to make accurate predictions in terms of diagnostics, machine learning algorithms need a significant amount of training datasets, which is difficult to access and varies per country due to their independence and legislation around data.

Understanding emotions and facial expressions are essential for social interaction, and autistic people struggle with this. Mobile apps aim to enable people with ASD to use their real potential and possibly overcome the aforementioned weaknesses [50]. By using AI-based algorithms to analyse the learners’ responses through an app, there is potential to discover significant insights into these individuals’ learning patterns and characteristics. In addition, online teaching and e-learning methodologies can assess learning progress while engaging learners.

### 5.1. Process Tracking and Personalised Content Delivery

The increasing partnerships in the health sector between clinicians and data technologists, reinforced by the rising power of clinical informatics, are now creating new openings to yield productive results. With this change, there is an increase in the required skills to understand big data and the insights that can be drawn from various information collected from patients [51]. AI-powered technology can be implemented in mobile apps to track the behaviours and analyse the routines of mobile users having ASD. These could then help understand where the user is performing best and where he/she needs help in improving specific skills. For example, TrackAI, a project developed by Huawei, utilises deep learning and the science of teaching programs to recognise patterns [52]. The project unleashes the potential to measure visual functions and analyse live data in order to assess whether a child has a visual impairment so that necessary actions can be taken to correct the child’s visual acuity [52]. Deep Learning can thus be used in a similar way to curate personalised content as per the ASD child’s point of interest, for example, in a game app or an educational app. By monitoring the interests and choices of ASD children and incorporating them into a deep learning algorithm, mobile apps can make recommendations. These recommendations will be based on their identified patterns, with a goal to suggest additional materials for improving upon their weaknesses, which would stimulate brain development and other skills from a young age. Another example of a useful application is the Time app, which uses AI and statistics to help users limit procrastination and complete their goals ahead of deadlines [53]. The app learns the working patterns of each user in a given scenario and subsequently provides recommendations on ways to increase productivity.

Furthermore, the app presents a visualisation functionality, which provides insights to the users on how much time they have worked, saved and added every hour. Therefore, similar concepts can be employed in apps designed explicitly for ASD people. The augmented features adapted from the Timeapp will enable ASD people and their informal carers to set specific goals and maintain logs, respectively, identifying the completed tasks and those that required more time.

### 5.2. Automated Reasoning

Automated reasoning is more complex than just analysing mobile users’ behaviours since it requires problem-solving based on numerous processes. For example, combining both eye-tracking software and deep neural networks in mobile apps can help in the early diagnosis of ASD in children through automated reasoning. The lack of clinical resources for early ASD detection has been a long-lasting issue. Thanks to the advancement in computer vision techniques, game apps can generate data to improve behavioural coding in ASD therapies. This can be achieved through facial expressions (e.g., visual attention, eye gaze, eye contact, smile gestures, etc.), captured through an in-built camera in a smartphone [54]. Their responses can help clinicians to reduce the delay in diagnosis. The early detection of ASD signs through behavioural observation may also be improved through Child–Robot Interaction (CRI) [54]. Researchers can enhance reasoning by drawing appropriate inferences to each situation. In other words, they need to understand how relevant implications can be sketched to offer credible solutions to a particular task. An example of a mobile app that uses automated reasoning is Uber, which optimises routes and enables users to reach their destinations faster after analysing previous data from Uber drivers operating on similar routes.

Similarly, if communication apps targeted for ASD people are integrated with such functionalities, automatic reasoning could be applied and the apps could, for instance, provide suggestions for a sentence by just selecting some pictures from the app. Another application of automated reasoning in mobile apps could be useful for mobility and transportation since the cognitive profile of people with ASD can lead to some difficulties during their journey. This might include communication limitations or the inability to use public transport safely, and very often, the need to be accompanied by a caregiver. Currently, transport apps that include transport planning and interaction with maps are designed mainly for users with no known disability, which therefore fails to provide simple interfaces for ASD people. Transport apps applying automated reasoning can be developed through a semantic web [55], which makes particular usage of heterogeneous data sources combined with problem-solving capabilities to make crucial decisions [56]. Machine Learning methods usually result in a system of weighted links between inputs and outputs. In contrast, the semantic modelling methodology relies on human-understandable illustrations of the concepts, associations and rules that comprise the desired knowledge domain [56]. Hybrid apps should exploit the capability of automated reasoning within semantic web development for decision-making.

### 5.3. Image Recognition and Natural Language Processing

Although there is a growing body of research indicating that ASD children reveal deficits in the capability to recognise facial identities and expressions [57], the use of computer-based interventions (CBI) to improve social and emotional skills has proven to have some advantages for these individuals [58]. Facial Recognition innovation can facilitate facial processing, enabling ASD children to familiarise themselves with people they frequently interact with through their mobile devices. Facially expressed emotions are difficult for people with ASD to recognise; however, facial emotion recognition can be shown to children through different virtual characters in certain games. This could contribute significantly to children’s development of communication skills, as suggested by some studies [59]. However, certain emotions such as jealousy or guilt are more nuanced than others (e.g., happiness, sadness, anger), which are complex and cannot be taught simply by using avatars. AI can be used by ASD children to identify objects and create a coherent caption with proper sentence structure in the same way that an individual without any disability would write. Deep learning and NLP have already delivered accuracy in both image and digit classification [60,61,62,63,64]. In the future, we need more mobile apps that can use the enhanced automatic translation of text or digits to develop the communication skills of those with ASD.

### 5.4. Learning Re-Enforcement through Intelligent Tutoring Systems

ASD children face many challenges in traditional school environments, especially learning disabilities in regular classes. An intelligent tutoring system (ITS) is any computer system or software application that tries to replicate the performance of a human tutor by supporting the theory of learning. ITS provides direct and customised instructions while performing a task without human intervention within a problem domain such as mathematics, medical diagnosis and distance learning. The research community is extensively working on ITS to support one-to-one and collaborative learning. Some of the studies show that ITS supports ASD children in the learning process [34]. However, ITS is still yet to assist and supplement ASD children with better learning strategies through a mobile application.

Adapting ITS for mobile devices would appeal to ASD children in early year education and expand the possibility of collaborative learning. As a result, ITS enabled on mobile apps with different learning strategies and scaffolding techniques can be very helpful for ASD children to learn in a stress-free environment.

## 6. Conclusions

People with autism experience the world in a different way from neuro-typical people. Many of them struggle to form part of society while developing social interactions. The COVID-19 pandemic has adversely affected these people by having a huge impact on the diagnosis, treatment and prognosis of autism among individuals. The constantly changing social distancing measures along with new ways of interaction and a new ‘normal’ have led to the changes in their conventional treatment patterns of the pre-COVID-19 era, thus making it difficult for families to manage such individuals. Nevertheless, the use of telehealth solutions such as apps available in mobile marketplaces could potentially help these individuals and their families manage their everyday lives, leading to an improved QoL. However, with over 2 million mobile apps on the market, out of which many have not been evaluated, it is difficult to recognise the efficacy of these apps.

In this study, a fairly large set of mobile apps (i.e., 250) have been retrieved using appropriate keywords. Among 250 apps, 46 were identified after filtering out irrelevant apps based on defined elimination criteria such as ASD common users, medical staff, and non-medically trained people who interact with people with ASD. In order to review common functionalities and features, 25 apps were downloaded and analysed. The analysis of mobile apps for people with ASD has led this research to conclude that the important features from different categories for mobile apps development are based on eye-tracking, facial expression analysis, use of 3D cartoons, haptic feedback, engaging interface, text-to-speech, use of ABA, and AAC techniques. These features need to be considered by the software development industry when implementing apps for people with learning disabilities, particularly autistic children. However, there were few published materials from the literature regarding the identified apps from the Google Play store and their use of AI techniques as potential solutions. Following are some of the major domains where AI could potentially contribute to improving the quality of experience: (1) Progress Tracking; (2) Personalised Content Delivery; (3) Automated Reasoning; (4) Image Recognition, and (5) Natural Language Processing.

Game app data could be exploited to see how autistic children react in different situations and then provide similar scenarios to follow their interactions with the system. When it comes to educational games, AI techniques could be implemented to monitor performance and body language to identify patterns in eye movement and facial expressions automatically. It was noticed that apps employing AI were mostly communication apps, which utilise NLP and text-to-speech software to enhance communication skills.

The majority of young people with a learning disability are capable of paid employment if they are well prepared and receive the right support. AI-enabled apps play a vital role in our fast-paced modern society. However, challenges such as ethics and security need to be considered before building any sophisticated program to help people with ASD. In addition, facilitating academic and mental health resilience in autistic people is still a challenge. This study provided a feature analysis and future recommendations, which need to be considered to enable ASD people to enjoy an improved QoL with more independence.

## 7. Limitation and Future Research Directions

The study was conducted only on apps that were added to the Google Play Store; thus, apps identified were Android-based. Further study can be undertaken in the future to identify iOS-based apps from the Apple store. Mobile web-based apps developed for autistic people were not used for this study. The actual testing of the apps mentioned above on ASD participants was not possible due to the COVID-19 situation. It is important to ensure that research participants are not subjected to harm in any way [65]. In this study, we did not recruit participants; hence, no ethical approval was required. A detailed study can be carried out by recruiting ASD participants to record ASD supportive applications’ effectiveness and recommendations for ASD apps development. As part of our future work, we will seek ethical approval to recruit participants in order to safeguard participants’ welfare from the onset of the study.

## Figures and Tables

**Figure 1 diagnostics-11-01923-f001:**
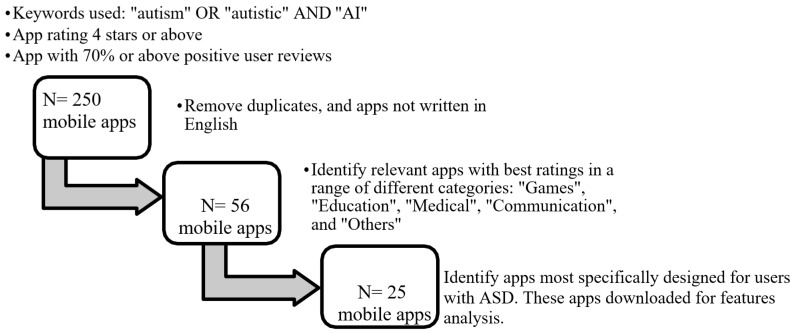
Mobile app selection process.

**Figure 2 diagnostics-11-01923-f002:**
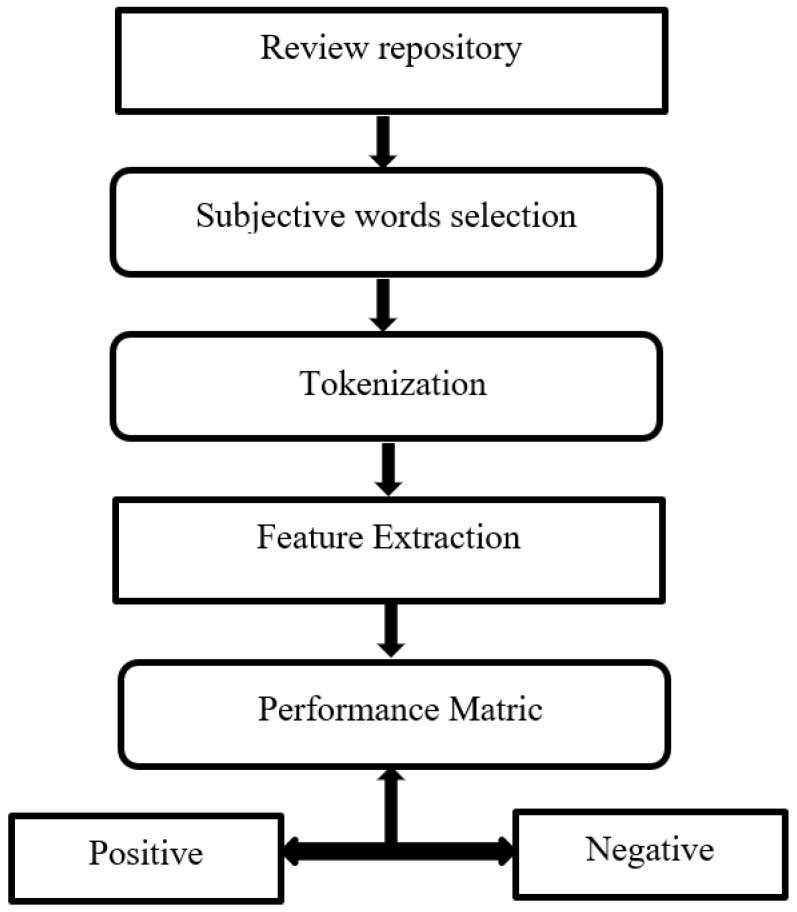
User reviews sentiment analysis.

**Figure 3 diagnostics-11-01923-f003:**
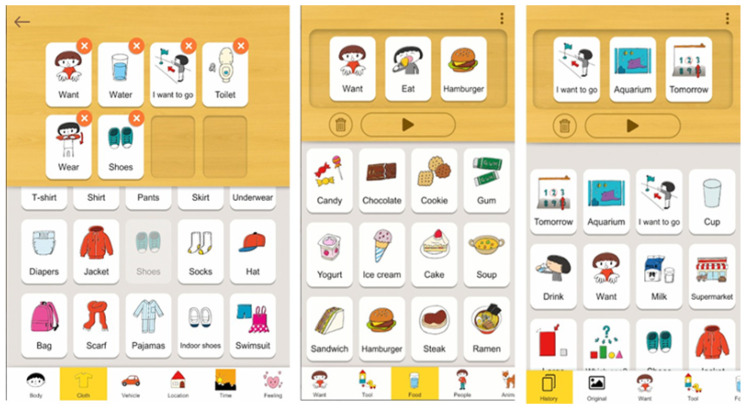
Card Talk App [39].

**Figure 4 diagnostics-11-01923-f004:**
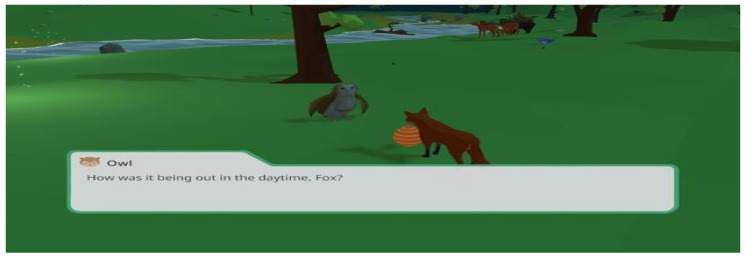
Prism App [40].

**Figure 5 diagnostics-11-01923-f005:**
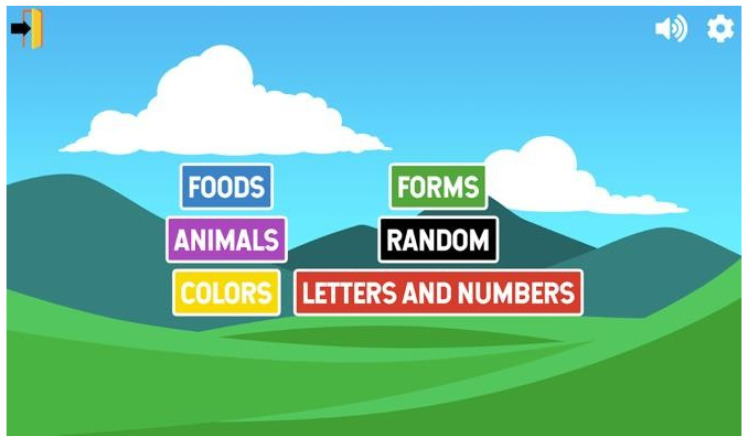
Jade Autism App [35].

**Figure 6 diagnostics-11-01923-f006:**
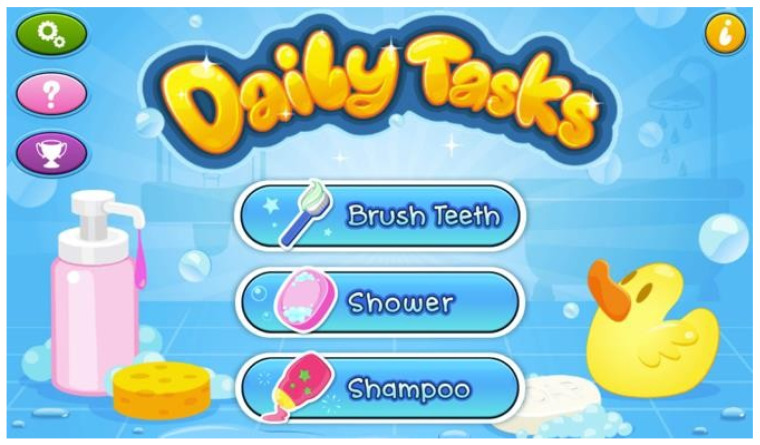
Daily Tasks App [36].

**Figure 7 diagnostics-11-01923-f007:**
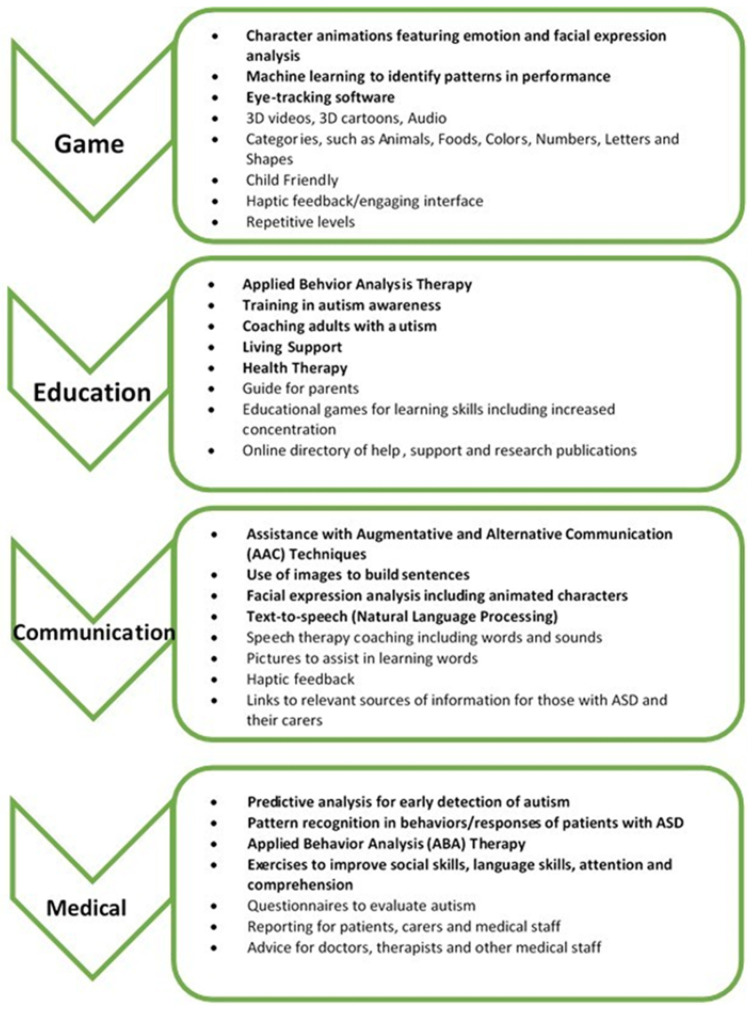
Features identified from the literature and mobile store.

**Table 1 diagnostics-11-01923-t001:** List of selected apps.

No	App Name	Description/Features	Rating (out of 5)	Category
1	Autism Language and Cognitive Therapy with MITA	Regular therapy, cognitive exercises for autistic children	4.6	Game
2	Jade Autism	Game for stimulating reasoning, cognitive and logical development in autistic children	4.6	Game
3	OTSIMO: Special education ABA therapy autism game	Game offering core skills (e.g., words, alphabets, numbers, emotions etc., through assistive matching, and drawing	4.0	Game
4	TEAPP: Autism and video games	Video game that promotes learning with special emphasis on the playful factor and fun in a 3D world	5.0	Game
5	Prism	A 3D open-world game for exploring colourful animal friends, enabling players to understand characteristics and behaviours commonly associated with autism	5.0	Game
6	Austism iHelp - Play	Real-life photographs presented in an easy-to-label format for enhanced learning	4.0	Education
7	ABA KIT	Teaching materials kit used in Applied Behaviour Therapy developed by ABA specialists	4.3	Education
8	Daily Tasks	Engaging and interactive app to improve eye-hand coordination, concentration and focus	4.3	Education
9	ASDetect	App for parents and caregivers to assess the social attention and communication behaviours of their children	4.3	Education
10	Understanding Autism	Short videos that are designed to enable caregivers to support people with ASD	5.0	Education
11	Awesomely Autistic Test	Short questionnaire to provide initial screening used by doctors before referring patients for a detailed formal evaluation	4.2	Medical
12	Autism Test	Test to help analyse the level of autism, providing instant results	4.0	Medical
13	ABA Therapy Aid	App with supported methods in ABA for the clinical, educational, and behavioural therapy of autism	4.0	Medical
14	Speech Assistant AAC	Medical app developed for people with ASD, enabling them to create categories, phrases and messages using speech-to-text technology	4.0	Medical/ Communication
15	ABA DrOmnibus for parents	Support tool including ABA principles for parents, teachers and therapists that help children to acquire the desired behaviour and new skills	4.5	Medical/ Education
16	LetMeTalk: Free AAC Talker	App developed to enhance communication skills among children with ASD (e.g., articulation/phonological problems, enabling users to line up images as a sentence	4.4	Communication
17	Card Talk	Card Talk supports communication with cards for children who have difficulties in verbal communcation	4.6	Communication
18	CommBoards Lite - AAC Speech Assistant	App to enable individuals express themselves more easily and communicate with the world around them, utilising a proven system of communication boards	4.3	Communication
19	JABtalk	App designed to help non-verbal children and adults communicate	4.0	Communication
20	Speech Therapy Articulation App 1 and 2 (UK)	App designed to help people with speech difficulties. The app consists of visual cues, games and fun exercises	4.8	Communication
21	The Autism Directory	App providing support and advice to individuals, families and professionals living and working with autism to lead an independent life	5.0	Others (Health/Fitness)
22	Moving Meditations for Kids with Autism	App, which includes videos that provide visual and movement experiences to help with stress related issues	5.0	Others (Health)
23	TIIMO: ADHD Autism App for Visual Structure	App for young adults/adults with special needs caused cognitive deficits. It combines a calendar, timer, icons and reminders in a single smart digital solution	4.4	Others (Lifestyle)
24	Autism Connect	App to connect autistic entrepreneurs with wider society	5.0	Others (Events/Social)
25	Autism NSW	App to help caregivers of ASD individuals in New South Wales to access all relevant organisation, support and points of contact	4.8	Others (Education/Social)

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
