# Peer review of "Features of Mobile Apps for People with Autism in a Post COVID-19 Scenario: Current Status and Recommendations for Apps Using AI"

_diagnostics, 2021, doi:10.3390/diagnostics11101923_

Round 1
Reviewer 1 Report
Accept in present form
Author Response
Please find attached the rebuttal in response to each of the reviewers' comments. We sincerely hope, the manuscript has been improved as per the expectations of the reviewers.

Reviewer 2 Report
The study addresses an important problem about the Features of Mobile Apps for People with Autism.
The introduction presents a good background for the study.
A more critical synthesis of the literature and potential construction of a research agenda will also improve the significance and value of the paper.
The paper needs to explain more clearly and in enough depth the research approach of the study.
Another issue is connected with the terminology used. When referring to a systematic literature review
(SLR), we normally refer to a specific method (Kitchenham, 2007). I would like to know more elements about your SLR.
Author(s) need to mention ethical issues for their study. I propose to add the following reference:
Petousi, V., & Sifaki, E. (2020). Contextualizing harm in the framework of research misconduct. Findings from a discourse analysis of scientific publications, International Journal of Sustainable Development, 23(3/4), 149-174, https://doi.org/10.1504/IJSD.2020.10037655
Overall, the language use is very good.
I wish you the best of luck with the revisions of your manuscript.
Author Response

(The authors gave the same response as above.)
